# Demographics, culture and participatory nature of multi-marathoning—An observational study highlighting issues with recommendations

Leo Lundy[1,2]*, Richard B. Reilly[1,2,3]

1 Trinity Department of Mechanical, Manufacturing and Biomedical Engineering, Trinity College, The University of Dublin, Dublin, Ireland, 2 School of Engineering, Trinity College, The University of Dublin, Dublin, Ireland, 3 School of Medicine, Trinity College, The University of Dublin, Dublin, Ireland

* lundyl@tcd.ie

**Data Availability Statement:** All relevant data are within the manuscript and its Supporting information files.

## Abstract

### Objectives

The defining achievement of a multi-marathoner is completing 100 marathons. This study aimed to comprehensively document the phenomenon of multi-marathoning, addressing its demographics, culture and participatory nature, filling a gap in peer-reviewed research on the topic. Additionally, it aimed to provide recommendations for multi-marathon governing bodies, event organisers, health professionals and participants to address identified issues.

### Methods

A global survey was distributed to participants and individuals interested in multi-marathoning. It was distributed with support from major national and international multi-marathon clubs through their social media channels, email groups and newsletters. The survey was conducted anonymously and online.

### Results

The survey garnered responses from 830 participants across 40 countries, with an average marathon completion count of 146.54 (SD 201.83) per respondent. Gender distribution showed 60.69% men, 39.3% women and 0.1% gender variant/non-conforming. Respondents' average ages were 51.6 (SD 9.96) years for men, 48.83 (SD 9.15) years for women and 35.00 (SD 8.76) years for gender variant/non-conforming. As participants age, social and travel motivations surpass competitiveness. A majority (57%) of respondents had at least one contravention to the pre-participation screening questionnaire PARQ-+ and 67% reported taking pain relief medication around events. Notably, 93% of respondents reported multi-marathoning as beneficial for their mental health.

**Funding:** The author(s) received no specific funding for this work.

**Competing interests:** The authors have declared that no competing interests exist.

**Abbreviations:** FREC, Faculty of Health Sciences Research Ethics Committee; HR, Heart Rate; PAR-Q+, Physical Activity Readiness Questionnaire for Everyone.

## Discussion

Multi-marathoning accommodates older athletes, but a significant gender imbalance exists in participation levels. Long-term health implications warrant attention from governing bodies, event organisers, health professionals and participants alike. Multi-marathoners should seek medical advice before participation, utilise modern equipment for health monitoring and optimise training accordingly.

## Conclusion

Recommendations include encouraging diversity at events, ensuring event directors have well-resourced health plans and promoting participants' proactive health management before and during their involvement in the sport. This study not only advances our understanding of multi-marathoning as a sport but also contributes to theoretical frameworks such as SDT and HBM.

## Introduction

Multi-marathoning, which emerged from the running boom of the 1970s and 1980s, has seen a resurgence in recent decades, with a focus on participation and completing organised events rather than competitive running [1–4]. The goals of multi-marathoners centre around completing a certain number of long-distance events, making the sport more inclusive and appealing to a wide range of athletes. Reaching, or being on the journey to the goal of 100 marathon completions defines a multi-marathoner.

The governance of multi-marathoning is decentralised, with 31 national bodies overseeing the sport and providing club amenities (including vetting marathon completion numbers), maintaining event standards, and organising events. International niche clubs also play a role, focusing on unique achievements such as completing marathons in multiple countries, often organising events in unconventional locations, such as the Vatican. Overarching this is a global world ranking system of official marathon completions. In 2023 this world ranking which included members from all 31 national governing bodies listed and ranked 1192 individuals globally who had completed over 300 vetted official marathons. In addition, a full ecosystem of events companies dedicated to servicing multi-marathoners has emerged globally.

This study explores this phenomenon of multi-marathoning guided by several theoretical frameworks that provide a conceptual foundation for a deeper understanding of the sport. One such framework is Self-Determination Theory (SDT), which puts forward that individuals are driven by intrinsic and extrinsic motivations in their participation in sports [5]. In the context of multi-marathoning, SDT suggests that participants may be motivated by a combination of intrinsic factors, such as the desire for personal achievement, well-being and social engagement, as well as extrinsic factors, such as peer recognition within the multi-marathon community.

Additionally, the Health Belief Model (HBM) offers insights into individuals' health-related behaviours [6]. According to the HBM, individuals' health behaviours are influenced by their perceptions of health threats, the severity of potential consequences, the benefits of adopting preventive actions and the barriers to accepting such changes. In the context of multi-marathoning, the HBM framework suggests that participants' engagement with proactive health intervention, such as seeking medical advice and using advanced running equipment, may be

influenced by their perceptions of the personal risks associated with marathon running, as well as their perceived benefits of maintaining a healthy lifestyle.

By integrating these theoretical frameworks, this study aims to provide a comprehensive understanding of the factors influencing multi-marathoners' engagement, behaviours and health outcomes. Through an analysis of socio-demographic, physiological and environmental factors, this research seeks to explore the relationships between individual characteristics, social influences and environmental factors in shaping the experiences of multi-marathoners.

As part of this study, an extensive literature review showed much research has been published on marathoning over recent decades. 'Marathoning' and associated keywords have over 12000 references on the 'PubMed' and 'Web of Science' archives of biomedical and life science literature alone, but on closer inspection, none of the peer-reviewed studies focused on or mentioned the sport of muti-marathoning. The absence of focused peer-reviewed research has resulted in a lack of awareness and understanding of the demographics, culture and participatory nature of multi-marathoning. This study aims to be the first to provide this understanding and address this gap in academic research, as well as understanding the motivations, behaviours, and health outcomes of participants.

Through careful selection of study variables, the study aims to investigate various facets of multi-marathoning and its associated factors. The research questions are structured as follows:

**Socio-demographic**: How do socio-demographic variables such as country location, age, gender identity, and dietary habits influence participants' involvement in multi-marathoning?

**Running History**: How do running history study variables like level of marathon completion, the distinction between ultra-marathon and regular marathon participation, annual event completion rate, performance and the age at which a participant began their long-distance running journey influence a participant's experience and longevity in the sport?

**Participation Motivation**: What intrinsic and extrinsic motivations drive individuals to participate in multi-marathoning? It was hypothesised that participants would value the social engagement multi-marathoning offered and that achieving 100 marathon completions would be their primary goal.

**Award Preferences**: What types of awards hold particular significance for participants, and how do these preferences vary across demographic groups and running histories?

**Participant Health**: How does participation in multi-marathoning influence participants' general health behaviours, particularly in the context of vigorous exercise pre-screening and the COVID-19 pandemic? Do multi-marathoners exhibit different patterns of pain medication usage compared to other athletes?

**Training Levels, Injuries Profiles and Recovery Techniques**: What are the characteristics of multi-marathoners' training regimens? What are the common injury profiles among multi-marathoners, and what recovery techniques do they employ to manage these injuries?

## Methods

Trinity College Dublin's (TCD) Faculty of Health Sciences Research Ethics Committee (FREC) ethically approved this study recommending that to minimise bias, any data collection instruments be distributed exclusively via gatekeepers with no direct approaches from the study team.

In line with the ethics committee recommendations, an online survey was developed to engage with the multi-marathoning running community globally. The survey was made

available on all common internet-enabled devices. There were 43 survey questions in total, that were grouped and categorised in line with the research question categories. The survey was open from Dec 14th, 2022, through to March 31st, 2023.

The survey was implemented using Qualtrics with a proxy server front end from Tiny URL giving a memorable survey custom URL https://multimarathon.study/survey [7].

Gatekeepers were identified by analysing the structure of multi-marathoning at a national and international level. All national and international multi-marathon clubs together with multi-marathon event companies in the UK and Ireland were approached by the study team. All the major multi-marathon clubs globally agreed to proactively support the distribution of the survey through their social media channels, email groups, or newsletters. The social media channels as well as engaging with the various club members were generally open to anyone interested in the sport.

In addition, all the major UK and Ireland multi-marathon event companies similarly supported the survey distribution to their social media channels which were open to anyone with an interest in the sport of multi-marathoning.

## Patient and public involvement

No patients or members of the public were involved in the design or interpretation of this study. All public participation was voluntary and informed consent was required before participation. Gatekeepers were exclusively used for the distribution of the survey the main results will be distributed via these gatekeepers to stakeholders.

## Equity, diversity and inclusion

The study included all genders, races/ethnicities, disabilities and socioeconomic levels.

# Results

The survey had 830 completed responses with a respondent average marathon completion count of 146.54 (SD 201.83).

## Socio-demographic characteristics

The survey yielded comprehensive socio-demographic data from the respondents, providing insights into the diverse composition of the multi-marathoning community. These socio-demographic characteristics underpin the rest of the results.

## Geographic distribution

Respondents came from 40 countries spanning 6 continents, indicating a global representation within the multi-marathoning community. The distributions of respondents varied across regions, with notable concentrations in regions known for their active running communities such as North America, Europe, and Oceania.

## Age distribution

The age distribution of respondents spanned a wide range, with individuals ranging from their twenties to their eighties. The average age for male participants was 51.6 years (SD 9.96), while female participants had an average age of 48.83 years (SD 9.15). Notably, participants in their sixties, seventies, and even eighties were actively engaged in multi-marathoning, highlighting the sport's appeal across different age groups.

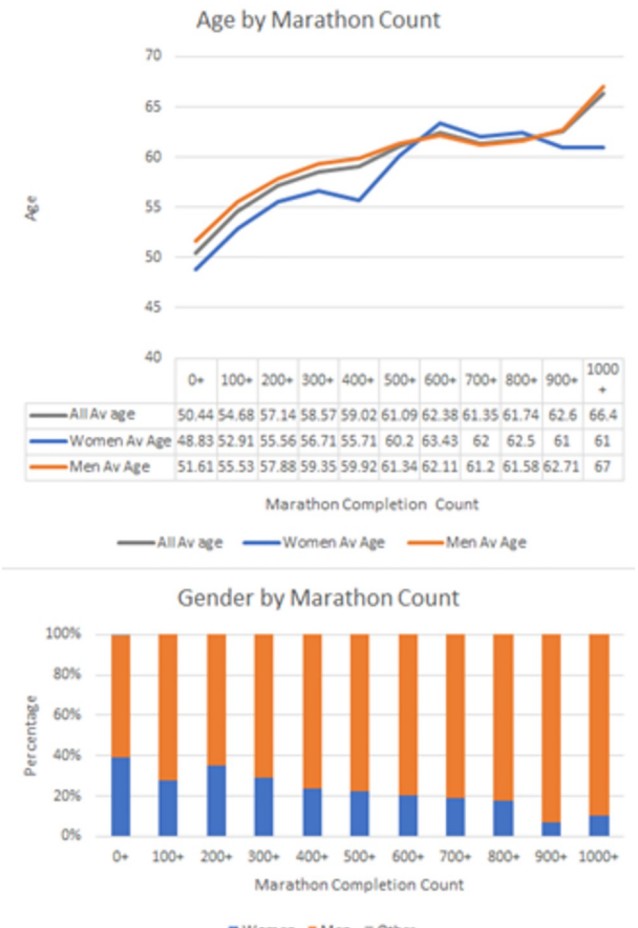

| | 0+ | 100+ | 200+ | 300+ | 400+ | 500+ | 600+ | 700+ | 800+ | 900+ | 1000+ |
|---|---|---|---|---|---|---|---|---|---|---|---|
| All Av age | 50.44 | 54.68 | 57.14 | 58.57 | 59.02 | 61.09 | 62.38 | 61.35 | 61.74 | 62.6 | 66.4 |
| Women Av Age | 48.83 | 52.91 | 55.56 | 56.71 | 55.71 | 60.2 | 63.43 | 62 | 62.5 | 61 | 61 |
| Men Av Age | 51.61 | 55.53 | 57.88 | 59.35 | 59.92 | 61.34 | 62.11 | 61.2 | 61.58 | 62.71 | 67 |

**Fig 1. Age and gender distribution of respondents.**

## Gender representation

The survey revealed a gender imbalance within the multi-marathoning community, with 60.69% of respondents identifying as male, 39.3% as female, and a small percentage (0.1%) identifying as gender variant/non-conforming (Fig 1). However, it is noteworthy that England exhibited a more balanced gender distribution, with 44.6% of respondents being male and 55.4% female.

## Dietary habits

Respondents provided insights into their dietary habits, with 69% indicating that they do not follow a specific diet (Fig 2). Among those who adhered to specific dietary preferences, significant numbers identified as vegetarian, vegan, paleo, or pesco-vegetarian. The prevalence of vegetarian diets was particularly noteworthy among participants.

This detailed socio-demographic information offers a nuanced understanding of the multi-marathoning community, encompassing geographical diversity, age demographics, gender representation, and dietary preferences. Such insights lay the foundation for further exploration into the motivations, behaviours, and health outcomes of multi-marathoners.

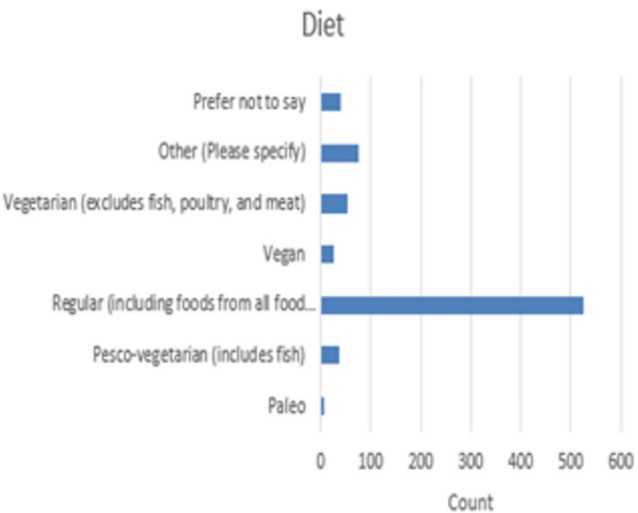

**Fig 2. Survey respondent's diet profile.**

## Running history and performance

For those with over 100 marathon completions on average 80% complete at least one marathon a month and on average 12% complete at least one marathon a week (Fig 3).

Reported performance showed most men having personal bests between 2:45–3:45 hrs., with a substantial number achieving less than 3:30 hrs. Most women reported having personal bests under 5:00 hrs., with a substantial number achieving less than 4:00 hrs.

Multi-marathoners reported that they run on average 57.91km (SD 9.8) a week including events. The effort was consistent throughout the year with many reporting that there was little peaking for specific events.

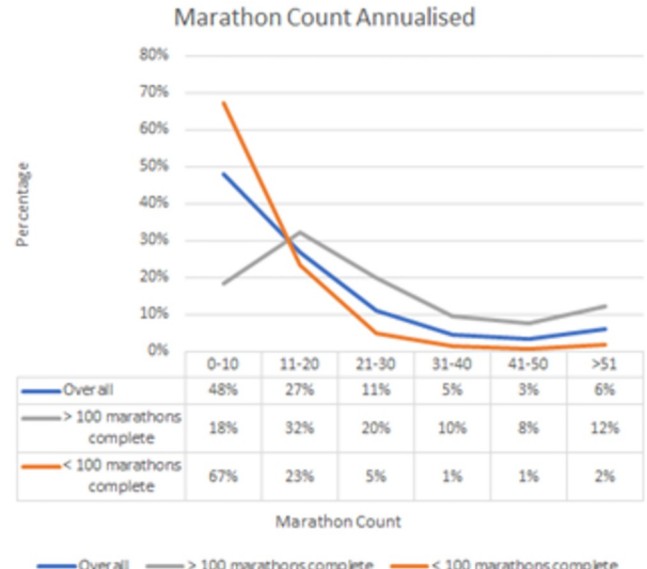

**Fig 3. Survey respondents marathon completion count by annualised frequency.**

## Motivations, awards and achievements

Most motivations detailed in the study were intrinsic, with keeping fit and healthy being the most popular (18.32%). A sense of accomplishment (16.26%) was closely followed by it being a way of life (13.62%). Social engagement was also seen as important (11.53%) with travel (8.82%) and reaching certain milestones (8.3%) also being significant.

Men with < 100 marathon completions reported being motivated by the need for continuous improvement and competition, but these motivations get replaced in the > 100 completions group where travel and social life take over as the main motivation.

Women prioritise social life, with reaching certain milestones being particularly important for women with < 100 marathon completions. Travel and social life were listed as more important in the women's group with > 100 marathons completions.

In terms of extrinsic motivations, most multi-marathoners reported they were members of multi-marathon awards clubs, with 45% of respondents reporting achieving 100 marathon completions and gaining the 100-marathon completion award from their national club as being the biggest single goal.

## Running statistics monitored

Steps, VO2$_{MAX}$ and Heart Rate (HR) parameters were the most popular statistics monitored by multi-marathoners. In general, multi-marathoners monitor whatever stats are available on their wrist-based GPS device and this varies from device to device.

## Equipment

From the survey, running equipment was important to multi-marathoners, with 93% reporting using GPS-enabled devices. Through these devices wrist-based HR monitoring was commonplace, but only 19% reported wearing the more accurate heart rate chest Strap. Additionally, only 22% said they use heart rate zones despite this information being readily available [8–10].

For training runs, 44% of multi-marathons said they use headphones regularly (or events when allowed). With the introduction of bone-conducting headphones, multi-marathoners have been at the forefront of their uptake with 41% of those who wear headphones reporting using this new headphone technology.

Cushioned and stable running shoes are the stalwart of the multi-marathoning community, with 45% of multi-marathoners reporting using cushioned shoes and 27% using stable shoes. Additionally, 20% reported preferring lightweight or minimal shoes (Fig 4).

## Injuries, treatments and recovery

The most common injuries reported were those associated with event-based issues blistering and chafing [11].

Other injuries suffered over the running career were common running injuries that affected joints (ankle, knee, hip) and muscular injuries (calf, muscle pulls) or tendon issues (Achilles, IT Band). Plantar Fasciitis was also commonplace [12,13]. No specific multi-marathoning injury types or specific overuse injuries were noted (Fig 5).

When injured, 50% of multi-marathoners reported attending physiotherapists to manage the injury with 23% reporting not seeking professional help at all. Treatments vary widely with self-medication (stretching, rest, RICE, foam rolling and pain relief) being stated as the most popular.

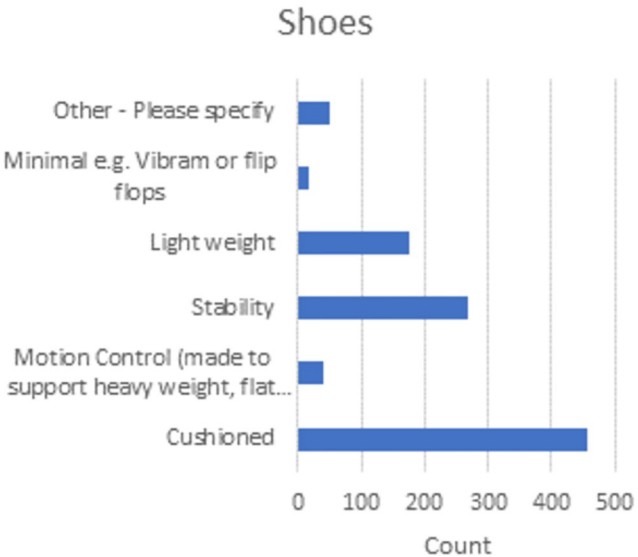

**Fig 4. Shoe types used by survey respondents.**

## Pain relief, mental health and life stressors

Despite much information and discussion regarding the dangers of taking pain relief medication during extreme sports, 67% of respondents say they take medications that relieve pain around events [14,15].

Respondents (15%) reported that their involvement in multi-marathoning had been a contributory factor to one of life's stressors (e.g., relationship breakdown, long-term illness, job loss).

More positively, 94% of respondents say that multi-marathoning was positive for their mental health.

## Health

The survey included the general health section of the PAR-Q+ questionnaire, an internationally recognised vigorous event pre-participation screening questionnaire. Any contraventions of this section of the PAR-Q+ questionnaire require athletes to answer further questions, seek further information, or see a health professional before partaking in vigorous exercise [4].

Based on the survey 57% of respondents reported that they had at least 1 contravention while 25% had 2 or more contraventions to the general health section of PAR-Q+. Of those respondents reporting contraventions 33% reported that they have bone, joint, or tendon issues that would get worse with continued running. Over 13% reported as to having high blood pressure and 10% reported having a heart condition. Significant numbers (9%) also reported having had dizziness or lost consciousness in the previous year with 17% reporting other chronic medical conditions.

Of the respondents who self-reported having had chronic medical conditions, 47% were taking prescribed medication to control this condition.

Covid was widespread in the community with 71% confirming they had Covid by way of a positive test or thought that they had Covid. Many reported that symptoms persisted after eight weeks including headaches, dizziness, tinnitus, septic arthritis, PoTS (Postural Tachycardia Syndrome), loss of taste and smell, excess perspiration, noise sensitivity, kidney pain,

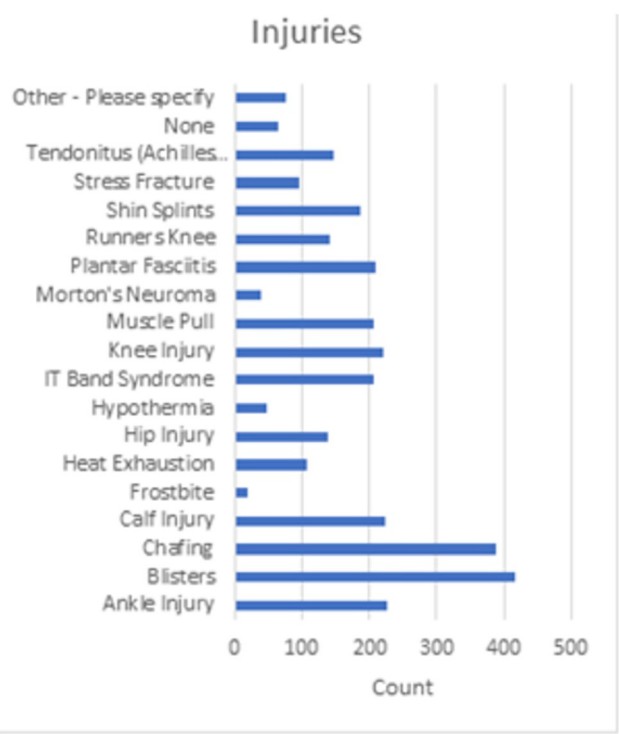

| | % | Count |
|---|---|---|
| Acupuncture | 3% | 77 |
| Compression | 8% | 190 |
| Elevation | 5% | 121 |
| Foam Rolling | 11% | 259 |
| Ice (Ice packs or cold compresses) | 10% | 235 |
| Pain Relief | 9% | 210 |
| Rest | 17% | 404 |
| RICE (Rest Ice Compression Elevation) | 8% | 179 |
| Support strapping | 5% | 131 |
| Stretching | 15% | 354 |
| Surgery | 1% | 25 |
| None | 2% | 46 |
| Other - Please Specify | 6% | 155 |

**Fig 5. Injuries and recovery techniques reported by survey respondents.**

cough, respiratory inflammation, memory recall, asthma, arrhythmia, toe numbness, excessive blood pressure variation, low blood pressure (Fig 6).

## Discussion

The socio-demographic data obtained from the survey provides valuable insights into the diverse composition of the multi-marathoning community and sheds light on the factors influencing participation, motivations, and experiences within the sport.

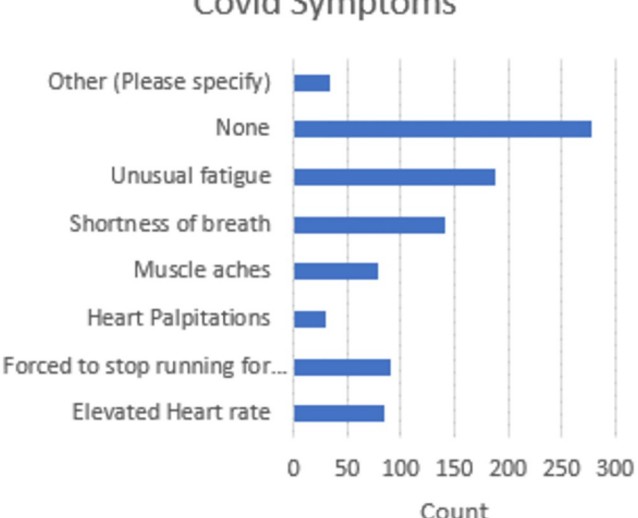

**Fig 6. Reported covid symptoms.**

The global distribution of participants reflects the widespread appeal of multi-marathoning across different regions and cultures. The concentration of respondents in regions with established running communities, such as North America, Europe, and Oceania, highlights the influence of local running cultures and infrastructures on participation rates. Future research could explore how cultural norms and societal attitudes towards endurance sports shape individuals' engagement with multi-marathoning in various geographical contexts.

The wide age range observed among participants highlights the inclusive nature of multi-marathoning, accommodating individuals from various life stages. The active participation of older individuals, including those in their sixties, seventies, and even eighties, challenges stereotypes associated with age. It suggests that multi-marathoning offers opportunities for life-long engagement in physical activity and underscores the importance of promoting inclusive practices within the sport to cater to diverse age groups.

The findings of this study offer valuable insights to various stakeholders within the multi-marathon community, including governing bodies, multi-marathon race directors, medical professionals and participants themselves.

From the perspective of governing bodies and event directors, the study emphasises the importance of promoting inclusivity and gender diversity within multi-marathon events. The gender imbalance observed in the multi-marathoning community warrants further exploration into the underlying factors influencing participation rates among different genders. While men constituted the majority of participants, the more balanced gender distribution in certain countries, such as England, suggests that cultural and societal factors may play a role in shaping gender dynamics within the sport. Future initiatives aimed at promoting gender equity and inclusivity in multi-marathoning could benefit from targeted interventions addressing barriers to participation and fostering a supportive environment for all genders.

Medical professionals are instrumental in promoting health awareness and preventive healthcare practices among multi-marathoners. The study reveals discrepancies between perceived health and actual health status among participants, underscoring the importance of regular health screenings and proactive engagement with medical resources. By advocating for adherence to established protocols and seeking professional medical advice when needed,

medical professionals can contribute to the well-being and longevity of multi-marathon participants.

Participants themselves offer unique perspectives on the motivations, challenges and aspirations associated with multi-marathoning. The study highlights the dedication of participants to achieving milestones such as 100 marathon completions. Additionally, the study highlights the significant frequency of marathon completion among those who have achieved this goal emphasising the enduring commitment and dedication of multi-marathoners to the sport.

Moreover, The study emphasises the significance of social connections as an intrinsic motivation within the multi-marathoning community, which becomes increasingly important with age. Understanding these evolving motivations can inform event organisers and governing bodies in tailoring initiatives to meet the changing needs of participants. Highlighted as well is the emphasis on social connections and the pursuit of personal fulfilment traverses different age groups and genders.

The prevalence of vegetarian and vegan diets among participants indicates the importance of dietary choices and health consciousness within the multi-marathoning community. Understanding the motivations behind dietary preferences, such as ethical concerns, health considerations, or performance optimisation, can inform strategies for promoting holistic well-being among multi-marathoners.

The study sheds light on multi-marathoners' adoption of running technology, with a preference for wrist-based devices with embedded heart rate monitoring capabilities. However, contrary to expectations, only a small percentage utilise heart rate zones to aid training. Recommendations from the study include leveraging heart rate training zones to optimise training and monitor heart health effectively.

Injuries within the sport primarily consist of typical running injuries, with no specific overuse multi-marathon injuries identified [12,13]. However, temporary event-related issues such as blisters and chafing remain common concerns [11]. The study highlights a significant percentage of participants not seeking medical advice for injuries, suggesting a need for increased awareness and proactive engagement with medical resources among multi-marathoners.

Despite the perceived benefits of multi-marathoning for mental health, a subset of participants reports it contributes to life stressors such as relationship breakdowns or long-term illness. Further research is warranted to explore the psychological impact of multi-marathoning and identify strategies to mitigate potential stressors.

The study also raises concerns regarding the use of pain relief medication around extreme events, with a significant proportion of respondents disregarding warnings against its usage. Recommendations emphasise the importance of avoiding pain relief medication without medical supervision to minimise associated risks [14,15].

Furthermore, the study highlights discrepancies between perceived health and actual health status among multi-marathoners, underscoring the importance of adhering to established protocols and seeking medical advice before engaging in vigorous exercise. The true state of health of multi-marathoners is not reflected in the information given to race directors when participants enter events. Subsequently, event directors do not have adequate equipment, training, or emergency planning to deal with significant incidents in events. Event directors need to understand this and plan accordingly.

Lastly, the impact of COVID-19 on multi-marathoners is noted, with a majority reporting contracting the virus and experiencing lingering symptoms. Further investigation is warranted to understand the long-term effects of COVID-19 on participation and performance levels among multi-marathoners.

These discussion points underscore the multifaceted nature of multi-marathoning and highlight key areas for further research and intervention to promote the well-being and longevity of participants in the sport.

## Limitations

While the study was designed to ensure respondent anonymity and minimise potential biases, several limitations should be acknowledged.

Firstly, the survey methodology of using gatekeepers with closed social media groups may introduce selection bias. Despite this being the most practical way of distributing the survey and the gatekeeper's social media channels being closed they were generally available not only to members of the gatekeeper clubs but also to anyone interested in multi-marathoning. However, this selection bias may impact the generalisability of the findings.

Secondly, the use of anonymous self-reported data, while practical for gathering information from participants, may introduce response bias. Respondents, while gaining no personal advantage, may be inclined to provide socially desirable or inaccurate responses, impacting the validity of the findings.

Also, the study sample is inherently biased towards individuals interested in and motivated by multi-marathoning. This selection bias excludes individuals who are unaware of the sport or its administering organisations, potentially limiting the generalisability of the findings.

Additionally, the survey did not capture insights from individuals who, for unknown reasons, abandoned the pursuit of multi-marathoning. Their perspectives could provide valuable insights into factors influencing participation and retention within the sport.

Furthermore, the survey was only available in English, which may have limited participation from individuals from non-English-speaking regions. As a result, the study may not fully represent the experiences and perspectives of multi-marathoners from these regions. Further research that includes surveys in multiple languages is recommended to better cover these regions and capture a more diverse range of perspectives.

Despite these limitations, the findings of this study provide valuable insights into various aspects of multi-marathoning, laying the groundwork for future research and initiatives aimed at promoting the well-being and inclusivity of multi-marathoners worldwide.

## Research/Policy implications

Until now, the absence of peer-reviewed research has resulted in a lack of understanding of the demographics, culture and participatory nature of the sport.

Multi-marathon participants and those involved in multi-marathon governance, in the provision of multi-marathon events, or health professionals, may utilise the recommendations given in this study to better plan their contribution to the sport and its overall safety, policies and organisation.

## Conclusion

In conclusion, this study provides valuable insights into the multi-dimensional nature of multi-marathoning, encompassing demographic characteristics, motivations, health considerations, dietary patterns, equipment preferences and injury profiles. The findings shed light on the unique appeal of multi-marathoning, particularly among older athletes and highlight the significance of achieving milestones such as 100 marathon completions.

From the perspective of governing bodies, there is a clear imperative to prioritise initiatives that foster inclusivity and gender diversity within multi-marathon events.

Multi-marathon race directors play a pivotal role in shaping the event experience and the study highlights key factors that contribute to participant satisfaction and engagement. By leveraging insights from this study, race directors can tailor event offerings to meet the diverse needs and preferences of multi-marathoners, thereby enhancing the overall participant experience and fostering long-term engagement with the sport. In terms of safety, event directors need to understand the state of health of participants entering their events and plan accordingly.

Medical professionals play a crucial role in promoting health awareness and preventive healthcare practices among multi-marathoners. The study reinforces the importance of regular health screenings and proactive engagement with medical resources to ensure the well-being and longevity of participants. By advocating for adherence to established protocols and providing guidance on injury prevention and management, medical professionals can support multi-marathoners in pursuing their athletic goals safely and sustainably.

The conclusions of this study resonate with Self-Determination Theory (SDT), showcasing how multi-marathoning provides a platform for both intrinsic satisfactions such as personal achievements or social interactions and extrinsic rewards such as milestone completions and peer recognition.

Moreover, the gender disparities observed in multi-marathoning and the perceived success of gender-balanced events in England highlight the importance of environmental factors and social support in shaping participation patterns and lay a foundation for further research in this area.

The study's findings also emphasise the importance of health awareness and proactive engagement with medical resources among multi-marathoners, aligning with the Health Belief Model (HBM). By identifying discrepancies between perceived health and actual health status among multi-marathoners, this study emphasises the need for targeted interventions to enhance health awareness and promote preventive healthcare practices within the multi-marathoning community.

Ultimately, participants themselves are at the heart of the multi-marathoning community and their motivations, challenges and aspirations shape the evolution of the sport. The study highlights the enduring commitment of participants to achieving milestones and the importance of social connections in sustaining their engagement with multi-marathoning.

In conclusion, this study not only advances our understanding of multi-marathoning as a sport but also contributes to theoretical frameworks such as SDT and HBM. Moving forward, future research should continue to explore these dimensions and examine the effectiveness of targeted interventions in promoting the diversity, well-being and longevity of multi-marathoners.

## Supporting information

**S1 File.**
(XLSX)

**S2 File.**
(PDF)

## Acknowledgments

The authors of this study would like to offer thanks to all study participants for their contribution to this research.

## Author Contributions

**Conceptualization:** Leo Lundy.

**Data curation:** Leo Lundy.

**Formal analysis:** Leo Lundy.

**Investigation:** Leo Lundy.

**Methodology:** Leo Lundy.

**Project administration:** Leo Lundy.

**Supervision:** Richard B. Reilly.

**Validation:** Leo Lundy.

**Visualization:** Leo Lundy.

**Writing – original draft:** Leo Lundy.

**Writing – review & editing:** Richard B. Reilly.

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
