## [Decision Letter · Decision Letter 0]

14 Feb 2024

PONE-D-23-38311Demographics, culture, and participatory nature of multi-marathoning – an observational study highlighting issues with recommendations.PLOS ONE

Dear Dr. Lundy,

Thank you for submitting your manuscript to PLOS ONE. After careful consideration, we feel that it has merit but does not fully meet PLOS ONE’s publication criteria as it currently stands. Therefore, we invite you to submit a revised version of the manuscript that addresses the points raised during the review process.

We look forward to receiving your revised manuscript.

Kind regards,

Hamed Ahmadinia

Academic Editor

PLOS ONE

Journal Requirements:

Additional Editor Comments:

Dear Authors,

Thank you for considering Plos One for publishing your valuable study entitled "Demographics, culture, and participatory nature of multi-marathoning – an observational study highlighting issues with recommendations". Your study with manuscript number PONE-D-23-38311 has been reviewed by two reviewers, and after an in-depth examination of your manuscript, it has been concluded that extensive revision is crucial prior to any further contemplation for publication. Key areas for improvement include enhancing the introduction to better justify the study variables, providing more detailed sociodemographic data in the results, and deepening the analysis in the discussion section. Additionally, the presentation style needs revision, particularly in converting point-form information into text and adjusting sentences that begin with numbers.

Furthermore, your study would benefit significantly from a clearer theoretical framework, addressing potential biases in your methodology, and strengthening the connection between your findings and the recommendations proposed. Reviewer 2 particularly emphasises the importance of including a variety of perspectives to enrich the study's depth. This comprehensive approach will help in addressing the methodological and analytical limitations, thereby enhancing the academic rigour and practical implications of your research.

Please ensure that these revisions are thoroughly addressed in your resubmission. We look forward to reviewing your revised manuscript.

Best Regards,

Hamed Ahmadinia

Academic Editor

PLOS ONE

Reviewers' comments:

Reviewer's Responses to Questions

**Comments to the Author**

1. Is the manuscript technically sound, and do the data support the conclusions?

Reviewer #1: Partly

Reviewer #2: Partly

2. Has the statistical analysis been performed appropriately and rigorously? 

Reviewer #1: N/A

Reviewer #2: Yes

3. Have the authors made all data underlying the findings in their manuscript fully available?

Reviewer #1: Yes

Reviewer #2: Yes

4. Is the manuscript presented in an intelligible fashion and written in standard English?

Reviewer #1: Yes

Reviewer #2: Yes

5. Review Comments to the Author

Reviewer #1: Dear authors,

After reading your work, I consider essential to pay attention to some concerns associated with this paper.

The introduction section should be reinforced. This section fails to justify the inclusion of the variables to be analysed with this study.

Results, we need more info about sociodemographics characteristics of the subjects.

Discussion section should be more deep. It is essential to analyse, associate and justify the results.

There are more practical implications of these results the the presented.

The info that was presented in points, should be presented in text.

Some sentences start with numbers. Please fix it.

Reviewer #2: "Demographics, culture, and participatory nature of multi-marathoning" by Lundy provide valuable insights into the multi-marathoning sport, shedding light on the motivations, demographics, and health implications associated with this activity. However, several issues with the study need to be addressed.

Firstly, the study needs a clear theoretical framework or conceptual foundation. While it reports on multi-marathoning's demographics, culture, and participatory nature, it fails to situate these findings within existing theoretical perspectives or prior research. This omission weakens the study's academic rigour and limits its contribution to the broader understanding of multi-marathoning.

The paper briefly mentions the historical context of running in the 1970s and 1980s, and it needs a thorough review of existing peer-reviewed research on multi-marathoning. A more extensive literature review would provide a stronger theoretical foundation for the study and demonstrate the gaps in knowledge that the current research aims to address.

Secondly, the survey methodology used in the study may introduce selection bias. The survey was distributed globally to sports participants via closed social media and email groups of major national and international multi-marathon clubs. This method may have resulted in a sample representative of only some of the multi-marathoning population, as it primarily captures the perspectives of individuals already engaged with multi-marathoning clubs. The lack of random sampling and the reliance on self-selected respondents raise questions about the generalizability of the findings.

Additionally, the study's reliance on self-reported data from the survey respondents introduces the potential for response bias. Respondents may have provided socially desirable responses or inaccurate information, particularly regarding their health behaviours and motivations for participation. This compromises the reliability and validity of the study's results.

Furthermore, while well-intentioned, the findings must sufficiently support the study's recommendations. The paper suggests implementing women-specific enhancements to events, ensuring race directors have adequately resourced health plans at events, and encouraging participants to take accountability for their health. However, these recommendations are separate from the study's specific findings. They may benefit from a more robust data analysis to support their relevance and feasibility.

In conclusion, while the study provides valuable insights into multi-marathoning sports, addressing the above methodological and analytical limitations is essential to strengthen the research's academic rigour and practical implications.

The author should consider incorporating additional points of view in the research to enrich the study's depth and breadth. The study would benefit from including diverse perspectives to offer a more comprehensive understanding of the subject.

Incorporating multiple points of view would enhance the study's academic rigour and contribute to a more nuanced portrayal of the multi-marathoning phenomenon. By including perspectives from a broader range of stakeholders, such as athletes, event organisers, health professionals, and governing bodies, the research could capture a more holistic view of the complexities and implications associated with multi-marathoning.

Additionally, incorporating diverse viewpoints would allow for a more thorough exploration of the motivations, challenges, and experiences of multi-marathon participants and the perspectives of those involved in organising and regulating multi-marathon events. This multifaceted approach would enrich the study's findings and provide a more comprehensive foundation for the recommendations offered to governing bodies and event organisers.

By integrating multiple points of view, the study could offer a more balanced and inclusive analysis of the sport, addressing a broader array of concerns and considerations. This approach would strengthen the academic quality of the research and enhance its practical relevance and applicability within the multi-marathoning community.

Incorporating additional points of view in the research would enrich the study's depth and provide a more comprehensive understanding of multi-marathoning, benefiting both academic scholarship and the practical implications for stakeholders involved in the sport.

6. PLOS authors have the option to publish the peer review history of their article (what does this mean?). If published, this will include your full peer review and any attached files.

Reviewer #1: No

Reviewer #2: **Yes: **Hugo Vieira Pereira

---

## [Author Response · Author response to Decision Letter 0]

2 Apr 2024

Editor: PLoS One

Title of the article: Demographics, culture, and participatory nature of multi-marathoning – an observational study highlighting issues with recommendations.

Rebuttal document

Thank you for the comprehensive comments from the editor and reviewers for this submission. We can certainly understand why these comments were made and we have taken them all on board and have substantially rewritten the whole submission based on this feedback.

We believe that this study which is a documentation of a new sport is important not only to academics but also to the professionals and participants that are involved in the sport.

Journal requirements

This revision of the submission meets PLOS ONE’s style requirements as outlined by PLOS ONE.

Data Availability statement

The entire data set will be made freely accessible if the submission is accepted for publication. The data availability statement in the submission form will reflect this.

Author affiliation

All authors’ affiliations reflect the institution where the study was carried out.

Editors comments

Key areas for improvement include enhancing the introduction to better justify the study variables, providing more detailed sociodemographic data in the results, and deepening the analysis in the discussion section. Additionally, the presentation style needs revision, particularly in converting point-form information into text and adjusting sentences that begin with numbers.

Furthermore, your study would benefit significantly from a clearer theoretical framework, addressing potential biases in your methodology, and strengthening the connection between your findings and the recommendations proposed. Reviewer 2 particularly emphasises the importance of including a variety of perspectives to enrich the study's depth. This comprehensive approach will help in addressing the methodological and analytical limitations, thereby enhancing the academic rigour and practical implications of your research.

In this revised submission, we have attempted to compressively address all the editor's concerns: a strengthened introduction justifying study variables, detailed sociodemographic data in the results section, and a deepened analysis in the discussion section. The presentation style has been revised, adhering to the feedback received. We have also addressed each of the reviewers’ comments below, point by point and noting where text has been edited or added to the manuscript. We acknowledge potential biases and address them in the revised manuscript.

By incorporating these comments and recommendations, we believe the depth and rigour of our study have been enhanced; We thank again the Editor and Reviewers for their helpful comments and suggestions.

Reviewer 1 comments

The introduction section should be reinforced. This section fails to justify the inclusion of the variables to be analysed with this study. Results, we need more info about sociodemographic characteristics of the subjects. Discussion section should be more deep. It is essential to analyse, associate and justify the results. There are more practical implications of these results the presented.

The Introduction has been rewritten and strengthened and now includes evidence supporting all variables included in the study. Specifically, this includes the addition of the following lines of text.

Through careful selection of study variables, the study aims to investigate various facets of multi-marathoning and its associated factors. The research questions are structured as follows:

Socio-demographic: How do socio-demographic variables such as country location, age, gender identity, and dietary habits influence participants' involvement in multi-marathoning?

Running History: How do running history variables like level of marathon completion, the distinction between ultra-marathon and regular marathon participation, annual event completion rate, performance and the age at which a participant began their long-distance running journey influence a participant's experience and longevity in the sport?

Participation Motivation: What intrinsic and extrinsic motivations drive individuals to participate in multi-marathoning? It was hypothesised that participants would value the social engagement multi-marathoning offered and that achieving 100 marathon completions would be their primary goal.

Award Preferences: What types of awards hold particular significance for participants, and how do these preferences vary across demographic groups and running histories?

Participant Health: How does participation in multi-marathoning influence participants' general health behaviours, particularly in the context of vigorous exercise pre-screening and the COVID-19 pandemic? Do multi-marathoners exhibit different patterns of pain medication usage compared to other athletes?

Training Levels, Injuries Profiles and Recovery Techniques: What are the characteristics of multi-marathoners' training regimens? What are the common injury profiles among multi-marathoners, and what recovery techniques do they employ to manage these injuries?

The results section has been significantly expanded to comprehensively include and contextualise sociodemographic characteristics. All results are now presented with respect to the study's sociodemographic characteristics. 

Specifically, this includes the addition of the following lines of text.

Results

The survey had 830 completed responses with a respondent average marathon completion count of 146.54 (SD 201.83).

Socio-Demographic Characteristics

The survey yielded comprehensive socio-demographic data from the respondents, providing insights into the diverse composition of the multi-marathoning community. These socio-demographic characteristics underpin the rest of the results.

Geographic Distribution

Respondents came from 40 countries spanning across 6 continents, indicating a global representation within the multi-marathoning community. The distributions of respondents varied across regions, with notable concentrations in regions known for their active running communities such as North America, Europe, and Oceania. 

Age Distribution

The age distribution of respondents spanned a wide range, with individuals ranging from their twenties to their eighties. The average age for male participants was 51.6 years (SD 9.96), while female participants had an average age of 48.83 years (SD 9.15). Notably, participants in their sixties, seventies, and even eighties were actively engaged in multi-marathoning, highlighting the sport's appeal across different age groups.

Gender Representation

The survey revealed a gender imbalance within the multi-marathoning community, with 60.69% of respondents identifying as male, 39.3% as female, and a small percentage (0.1%) identifying as gender variant/non-conforming. However, it is noteworthy that England exhibited a more balanced gender distribution, with 44.6% of respondents being male and 55.4% female.

Dietary Habits

Respondents provided insights into their dietary habits, with 69% indicating that they do not follow a specific diet. Among those who adhered to specific dietary preferences, significant numbers identified as vegetarian, vegan, paleo, or pesco-vegetarian. The prevalence of vegetarian diets was particularly noteworthy among participants.

This detailed socio-demographic information offers a nuanced understanding of the multi-marathoning community, encompassing geographical diversity, age demographics, gender representation, and dietary preferences. Such insights lay the foundation for further exploration into the motivations, behaviours, and health outcomes of multi-marathoners.

We have comprehensively rewritten the Discussion section which now provides a deep interpretation of the study's implications and now align the findings of the study and recommendations with real-world practicalities. Rather than make speculative recommendations, not directly related to the results, all recommendations come directly from the results or form a basis for further research and are detailed as such. We have also significantly expanded the Limitations sections to reflect the reviewer's recommendations.

The info that was presented in points, should be presented in text. Some sentences start with numbers.

This point has been addressed in this revision of the submission. In the Results and Discussion sections, we have rewritten the points in longer, clearer sentences and corrected any sentences starting with numerals. 

Reviewer 2 comments

The study needs a clear theoretical framework or conceptual foundation. While it reports on multi-marathoning's demographics, culture, and participatory nature, it fails to situate these findings within existing theoretical perspectives or prior research. This omission weakens the study's academic rigour and limits its contribution to the broader understanding of multi-marathoning.

The Introduction, Discussion and Conclusion sections have been rewritten and are now fully integrated with two relevant and well-known theoretical frameworks and the study is now fully situated relative to the perspectives these frameworks bring. These theoretical frameworks are the Self-determination Theory (SDT) and the Health Belief Model (HMB). We agree with the reviewer that the inclusion of a clear theoretical framework perspective much enhances the rigour of the study and adds to the body of work for these theoretical frameworks.

The paper briefly mentions the historical context of running in the 1970s and 1980s, and it needs a thorough review of existing peer-reviewed research on multi-marathoning. A more extensive literature review would provide a stronger theoretical foundation for the study and demonstrate the gaps in knowledge that the current research aims to address.

The original submission in the Summary section mentions an extensive literature review done as part of this study, where it describes that despite over 12000 references to marathon running in biomedical and life science literature, none references or mentions the sport of multi-marathoning. 

The introduction section now expands on this and introduces this literature review and its findings and emphasises that based on this literature review the main aim of the study was to address the lack of peer-reviewed research that this literature review uncovered.

As part of this study, an extensive literature review showed much research has been published on marathoning over recent decades. ‘Marathoning’ and associated keywords have over 12000 references on the ‘PubMed’ and ‘Web of Science’ archives of biomedical and life science literature alone, but on closer inspection, none of the peer-reviewed studies focused on or mentioned the sport of muti-marathoning. The absence of focused peer-reviewed research has resulted in a lack of awareness and understanding of the demographics, culture and participatory nature of multi-marathoning. This study aims to be the first to provide this understanding and address this gap in academic research.

The survey methodology used in the study may introduce selection bias. The survey was distributed globally to sports participants via closed social media and email groups of major national and international multi-marathon clubs. This method may have resulted in a sample representative of only some of the multi-marathoning population, as it primarily captures the perspectives of individuals already engaged with multi-marathoning clubs. The lack of random sampling and the reliance on self-selected respondents raise questions about the generalizability of the findings.

The gatekeepers were chosen carefully for this study and included the national governing bodies that are responsible for the administration of the sport. Even though these bodies do operate closed social media groups, these groups are open to anyone who has an interest in the sport and not just members. By way of example, in Ireland the governing body of multi-marathoning, Marathon Club Ireland has 199 official club members, but its social media group on Facebook services over 4300 interested athletes, which would account for those in the country with even a passing interest in the sport of multi-marathoning. This is mirrored across the globe. However, we recognise that this methodology may introduce a selection bias as described, from those athletes only engaged with multi-marathon clubs' social media. A statement reflecting this bias has been included in the submission.

.. the survey methodology of using gatekeepers with closed social media groups may introduce selection bias. Despite this being the most practical way of distributing the survey and the gatekeeper’s social media channels being closed, they were generally available not only to members of the gatekeeper clubs but also to anyone interested in multi-marathoning. However, this selection bias may impact the generalisability of the findings.

Additionally, the study's reliance on self-reported data from the survey respondents introduces the potential for response bias. Respondents may have provided socially desirable responses or inaccurate information, particularly regarding their health behaviours and motivations for participation. This compromises the reliability and validity of the study's results.

The study has been designed to ensure respondent anonymity, with no incentive or advantage for socially desirable or inaccurate responses. We believe that self-reported data was the most feasible method for gathering this type of information from participants and that this approach aligns with standard practices for studies of this nature. However, recognising the possibility of response bias as described by the reviewer, we have incorporated a statement addressing this bias in the Limitations section of the submission.

…. the use of anonymous self-reported data, while practical for gathering information from participants, may introduce response bias. Respondents, while gaining no personal advantage, may be inclined to provide socially desirable or inaccurate responses, impacting the validity of the findings.

While well-intentioned, the findings must sufficiently support the study's recommendations. The paper suggests implementing women-specific enhancements to events, ensuring race directors have adequately resourced health plans at events, and encouraging participants to take accountability for their health. However, these recommendations are separate from the study's specific findings. They may benefit from a more robust data analysis to support their relevance and feasibility. 

We appreciate the recommendation to enhance alignment between study findings and recommendations. In this revision of the submission, we ensure a concise and contextual presentation of recommendations directly tied to our study findings. Where a comment is not directly supported by the results, this has been highlighted, contextualised and described as such. 

The author should consider incorporating additional points of view in the research to enrich the study's depth and breadth. The study would benefit from including diverse perspectives to offer a more comprehensive understanding of the subject. Incorporating multiple points of view would enhance the study's academic rigour and contribute to a more nuanced portrayal of the multi-marathoning phenomenon. By including perspectives from a broader range of stakeholders, such as athletes, event organisers, health professionals, and governing bodies, the research could capture a more holistic view of the complexities and implications associated with multi-marathoning. Additionally, incorporating diverse viewpoints would allow for a more thorough exploration of the motivations, challenges, and experiences of multi-marathon participants and the perspectives of those involved in organising and regulating multi-marathon events. This multifaceted approach would enrich the study's findings and provide a more comprehensive foundation for the recommendations offered to governing bodies and event organisers. By integrating multiple points of view, the study could offer a more balanced and inclusive analysis of the sport, addressing a broader array of concerns and considerations. This approach would strengthen the academic quality of 

---

## [Editor Report · Decision Letter 1]

8 Apr 2024

Demographics, culture, and participatory nature of multi-marathoning – an observational study highlighting issues with recommendations.

PONE-D-23-38311R1

Dear Dr. Lundy,

We’re pleased to inform you that your manuscript has been judged scientifically suitable for publication and will be formally accepted for publication once it meets all outstanding technical requirements.

Kind regards,

Hamed Ahmadinia

Academic Editor

PLOS ONE